# Scattering of X-ray Ultrashort Pulses by Complex Polyatomic Structures

**DOI:** 10.3390/ijms23010163

**Published:** 2021-12-23

**Authors:** Dmitry Makarov, Anastasia Kharlamova

**Affiliations:** Northern (Arctic) Federal University, Laboratory of Diagnostics of Carbon Materials and Spin-Optical Phenomena in Wide-Gap Semiconductors, Nab. Severnoi Dviny 17, 163002 Arkhangelsk, Russia; a.a.harlamova@narfu.ru

**Keywords:** atoms, complex molecules, polyatomic structures, X-ray pulse, ultrashort pulse, scattering, X-ray diffraction analysis, DNA, RNA

## Abstract

The scattering of X-ray ultrashort pulses (USPs) is an important aspect of the diffraction analysis of matter using modern USP sources. The theoretical basis, which considers the specifics of the interaction of ultrashort pulses with complex polyatomic structures, is currently not well developed. In general, research is focused on the specifics of the interaction of ultrashort pulses with simple systems—these are atoms and simple molecules. In this work, a theory of scattering of X-ray ultrashort pulses by complex polyatomic structures is developed, considering the specifics of the interaction of ultrashort pulses with such a substance. The obtained expressions have a rather simple analytical form, which allows them to be used in diffraction analysis. As an example, it is shown that the obtained expressions can be used to study the structures of deoxyribonucleic (DNA) and ribonucleic (RNA) acids.

## 1. Introduction

X-ray ultrashort pulse (USP) scattering is the basis of X-ray diffraction analysis (XRD) [1,2,3,4]. In turn, XRD is one of the most important methods to study the structure and properties of matter. The structures of most crystals and many molecules have been determined using this method and underlie many modern discoveries in physics, chemistry, biology, medicine, and crystallography, such as [2]. Currently, XRD has many directions, which are complemented and expanded due to the creation of new types of radiation sources, increased power and decreased duration of ultra-short pulses, etc. [5,6,7,8,9].

Particular attention is now focused on ultrashort pulse physics [5,8,9]. Using such USPs, it is possible to conduct research on the structure of matter with high temporal and spatial resolution. This is even more so now that there is the technical possibility to conduct such studies. One of the most promising sources of USPs are free-electron XFELs [10]. At present, the formation of attosecond pulses is reported due to improvements in X-ray free-electron lasers techniques [11,12].

Researchers also reached the subfemtosecond barrier with high peak power, which allows the study of excitation in the molecular system and the movements of valence electrons with high temporal and spatial resolution, for example [13]. Due to the creation of high-power USP sources, there is a need for new theoretical approaches that take into account the specifics of the interactions of such USPs with complex polyatomic structures [6,14].

The theory of X-ray diffraction is typically constructed in the approximation of plane wave scattering of infinite time duration [15]. The scattering processes of femto- and particularly attosecond pulses on such structures have not been sufficiently studied and are actively developed in our time [6,16,17,18,19,20,21,22,23,24,25]. Usually such theories consider simple systems, such as atoms, simple molecules, model systems, systems of one-part atoms, etc. A simple theory that considers the specificity of X-ray scattering on complex polyatomic structures currently does not exist.

In the present paper, such a theory, obtained on the basis of the sudden perturbation approximation, will be presented. The results will have a simple analytical form and can be applied to calculations of scattering spectra for complex polyatomic structures. As an example, the case of scattering of X-ray USP on nucleotides (adenine, guanine, cytosine, thymine), which are the basis of deoxyribonucleic acid (DNA), is considered. It is shown that the scattering spectra are sensitive to spatial changes in the position of atoms in nucleotide structures. The results obtained can be easily extended to more complex structures, including deoxyribonucleic (DNA) and ribonucleic (RNA) acids.

Next, we will use the atomic system of units: *ℏ* = 1; |e| = 1; $m_{e}$ = 1, where *ℏ* is the Dirac constant, *e* is the electron charge, and $m_{e}$ is the electron mass.

## 2. Scattering of X-ray Ultrashort Pulses

We will assume that the USP falls in the direction of $n_{0}$ onto a complex polyatomic structure. We will assume that the time duration of such a pulse τ is much shorter than the characteristic atomic time $\tau_{a}∼1$, i.e., let us assume that $\tau \ll \tau_{a}$. This will allow us to use the sudden perturbation approximation. In this approximation, the eigen Hamiltonian of the system can be neglected, since the electron in the atom does not have time to evolve under the action of the USP field because the momentum interaction with the electron in the [26] atom is too fast.

It should be added that the condition $\tau \ll \tau_{a}$ to use our approximation is not strict. In the case of X-ray USPs, as was shown in [22,26], it is sufficient to assume that $\omega_{0}\tau_{a}\gg 1$, where $\omega_{0}$ is the carrier frequency of the incident USP. Let us choose the electromagnetic field strength of USP in general form $E\left(r,t\right)=E_{0}h\left(t-n_{0}r/c\right)$, i.e., we will consider spatially inhomogeneity, where $E_{0}$ is the field amplitude, and $h(t-n_{0}r/c)$ is an arbitrary function specifying the form of USP, *c* is the speed of light (in a.u. c≈137). In [26], when solving the Dirac equation, the wave function of the electron in the USP field with strength E(r,t) was found, which we will use.

We will consider the fields not so strong as to account for the magnetic field of the USP, i.e., we will assume that $E_{0}/c^{2}\ll 1$ or in units of intensity $I\ll 10^{25}\;W/{\mathrm{cm}}^{2}$. In this case, as shown in [26], the wave function of a complex multi-electron system can be represented as
(1)$$\begin{array}{c} \Psi \left(t\right)=\varphi_{0}\left(\left\{r_{a}\right\}\right)e^{-\sum_{a}i\int_{-\infty }^{t}E\left(r_{a},t^{\prime }\right)r_{a}dt^{\prime }}, \end{array}$$
where $\sum_{a}$ is the summation over all electrons in a complex polyatomic structures, $\varphi_{0}\left(\left\{r_{a}\right\}\right)$ is the initial wave function of all electrons in such a system.

To calculate the basic scattering characteristics, we will use the quantum theory of USP scattering, in which there are no restrictions on the number of atoms in the system [20]. In this theory, general expressions for calculations of the main scattering characteristics are derived. As a result, using Equation (Equation 1) and the theory in [20], we obtain an expression to calculate the scattering energy ε per unit solid angle $\Omega_{k}$ ($k=\frac{\omega }{c}n$, where n is the direction of the scattered pulse) in the unit frequency interval ω (hereafter the spectrum)
(2)$$\begin{array}{c} \frac{d^{2}\varepsilon }{d\Omega_{k}d\omega }=\frac{{\left[E_{0}n\right]}^{2}}{{\left(2\pi \right)}^{2}}\frac{|\tilde{h}{\left(\omega \right)|}^{2}}{c^{3}}\langle \varphi_{0}|\sum_{a,a^{{}^{\prime }}}e^{-ip(r_{a}-r_{a^{{}^{\prime }}})}|\varphi_{0}\rangle , \end{array}$$
where $\tilde{h}\left(\omega \right)=\int_{-\infty }^{+\infty }h\left(\eta \right)e^{i\omega \eta }d\eta$, and $p=\frac{\omega }{c}\left(n-n_{0}\right)$ has the meaning of recoil momentum when a USP is scattered on a bound electron. In Equation (Equation 2), the USP scattering spectrum is represented as an average over the ground state of a complex polyatomic system. To calculate the mean in Equation (Equation 2), one must know the wave function of the polyatomic structures and perform a multidimensional integration. It is clear that this problem cannot be solved directly for complex systems even considering modern computational power.

This problem can be solved using the electron density of an atom and considering complex polyatomic structures consisting of separate isolated atoms, i.e., using the model of independent atoms, see for example [22]. Dividing Equation (Equation 2) by two, where the first term corresponds to the summation at $a=a^{{}^{\prime }}$ and the second term at $a\neq a^{{}^{\prime }}$, we obtain
(3)$$\begin{array}{c} \frac{d^{2}\varepsilon }{d\Omega_{k}d\omega }=\frac{{\left[E_{0}n\right]}^{2}}{{\left(2\pi \right)}^{2}}\frac{|\tilde{h}{\left(\omega \right)|}^{2}}{c^{3}}\left[\sum_{i=1}^{A}N_{e,i}N_{A,i}\left(1-\right|F_{i}{|}^{2})+\sum_{i,j=1}^{A}\delta_{i,j}N_{e,i}N_{e,j}F_{i}F_{j}^{*}\right] \end{array}$$
where $N_{e,i}$ is the number of electrons in the atom *i* variety; $N_{A,i}$ is the number of atoms *i* variety; *A* is the number of varieties of atoms; and $F_{i}=\frac{1}{N_{e,i}}\int \rho_{e,i}\left(r\right)e^{-ipr}d^{3}r$ is the form factor of the *i* atom of the variety with electron density $\rho_{e,i}\left(r\right)$. The factor $\delta_{i,j}=\sum_{Ai,A^{{}^{\prime }}j}e^{-ip(R_{\mathrm{Ai}}-R_{A^{{}^{\prime }}j})}$ depends only on the coordinates of atoms *i* of the variety (with number Ai) whose position is determined by the radius vector $R_{Ai}$. Equation (Equation 3) is analytic, which contributes to a fairly simple calculation of the spectra. The main difficulty in the calculation is determined by the factor $\delta_{i,j}$, as, for complex systems, it is difficult to find an analytical expression for this.

This factor determines the interference, and, only in this factor, the coordinates of atoms in a complex polyatomic structures are concentrated. For fairly simple systems consisting of a single variety of atoms, such a factor has been found for many carbon systems [21,22]: graphene, nanotube, atomic rings, “forests” of nanotubes, etc. Equation (Equation 3) considers both coherent USP scattering (second term) and incoherent scattering (first term).

In the general case, the predominance of the coherent over the incoherent factor is determined by many factors. If the USP is multicycle, i.e., $\tau \omega_{0}\gg 1$ (τ is the pulse duration, $\omega_{0}$ is the carrier frequency of the USP), then scattering occurs at the frequency $\omega \to \omega_{0}$. In this case, coherent scattering dominates over incoherent scattering at $\lambda_{0}\gg 1$. In the case of multi-cycle USP, the predominance of the incoherent term over the coherent one, the problem is determined not only by the condition $\lambda_{0}\ll 1$ but also by the number of atoms in the system in question. In the case of low-cycle and sub-cycle pulses, it is necessary to consider a particular polyatomic structures and the form of the USP to determine the predominance of coherent over non coherent; this cannot be done in a general form.

If the polyatomic structure has a certain symmetry or periodicity, it is reflected only in the $\delta_{i,j}$ factor. For polyatomic systems, it is difficult to analyze and calculate the scattering spectra during the numerical calculation of Equation (Equation 3). In order to make the calculation and analysis simple, it is necessary for polyatomic structures with a certain symmetry and periodicity to represent the factor $\delta_{i,j}$ in the analytical form. In general, for such systems, the factor $\delta_{i,j}$ can be represented as
(4)$$\begin{array}{c} \delta_{i,j}=\sum_{\alpha =1}^{s}\sum_{n_{\alpha }=0}^{N_{\alpha }}e^{ipR_{n_{\alpha }}}\sum_{A_{i}\in R_{\alpha ,1}}e^{ipR_{A_{i}}}\sum_{\beta =1}^{s}\sum_{n_{\beta }=0}^{N_{\beta }}e^{-ipR_{n_{\beta }}}\sum_{A_{j}\in R_{\beta ,1}}e^{-ipR_{A_{j}}}\;, \end{array}$$
where α or β is some symmetry in the system, *s* is the number of symmetries in the system, $R_{n_{\alpha }}$ or $R_{n_{\beta }}$ is the radius vector that sets the symmetry position α or β respectively, $N_{\alpha }$ is the number of translations with a given symmetry, $R_{A_{i}}$ is the radius vector specifying the position of the atoms of variety *i* within the region $R_{\alpha ,1}$ (analogously $R_{A_{j}}$), see Figure 1.

The importance of Equation (Equation 4) is determined by the fact that it is not necessary to calculate numerically the parameter $\delta_{i,j}$ by summing up all positions of atoms in space. It is sufficient to determine the symmetry of the object under study and find the sum $\sum_{n_{\alpha }=0}^{N_{\alpha }}e^{ipR_{n_{\alpha }}}$ in analytical form. The sum $\sum_{A_{i}\in R_{\alpha ,1}}e^{ipR_{A_{i}}}$ can be found analytically if there is symmetry inside the region $A_{j}\in R_{\alpha ,1}$ or numerically, where summation is sufficient only in the region $R_{\alpha ,1}$.

For example, in the simplest case of a one-atom cubic lattice: s=α=1, i=j=const=1, $\sum_{A_{i}\in R_{\alpha ,1}}e^{ipR_{A_{i}}}=e^{ipR_{A_{1}}}$, since the atom within the symmetry is alone. As a result, for this case, the factor $\delta_{1,1}={\left|\sum_{n_{\alpha }=0}^{N_{\alpha }}e^{ipR_{n_{\alpha }}}\right|}^{2}$, which is well known [21] and calculated in a simple analytical form.

## 3. Scattering on DNA Nucleotides

The study of scattering on various polyatomic structures is a separate scientific task. The most interesting systems may be deoxyribonucleic (DNA) and ribonucleic (RNA) acids. Consider, for example, the DNA molecule. This molecule is based on nucleotides: adenine, guanine, cytosine, thymine. Each of the nucleotides is repeated in the DNA molecule, which means there is a symmetry that can be calculated in the factor $\delta_{i,j}$. The most interesting thing is that this symmetry can be modified by modeling the contraction, stretching or twisting of the DNA molecule.

The change in symmetry, and the symmetry itself, should be reflected in the scattering spectra calculated from Equation (Equation 3). In any case, knowledge of the scattering spectra on individual nucleotides is necessary when scattering a USP on such a molecule. Here, we will present such calculations. Let us calculate the scattering spectra on the following nucleotides separately: adenine, guanine, thymine, and cytosine. In this case, we need to find the factor $\delta_{i,j}$, with $s=1,N_{\alpha }=1$, then $\delta_{i,j}=\sum_{A_{i}\in R_{1,1}}e^{ipR_{A_{i}}}\sum_{A_{j}\in R_{1,1}}e^{-ipR_{A_{j}}}$. These nucleotides are non-periodic and asymmetric systems, and thus the calculation of the scattering spectrum will be conducted directly by substituting the coordinates of the atoms in the nucleotide into the factor $\delta_{i,j}$.

To calculate scattering spectra, we will use the model of independent atoms [27], in which molecules are represented by independent isolated atoms. The electron density of such atoms $\rho_{e,i}\left(r\right)=\frac{N_{e,i}}{4\pi r}\sum_{k=1}^{3}A_{k,i}\alpha_{k,i}^{2}e^{-\alpha_{k,i}r}$, where $A_{k,i},\alpha_{k,i}$ are constant coefficients (for all varieties of atoms with number *i*) defined in [27]. The result is a simple expression for $F_{i}=\sum_{k=1}^{3}\frac{A_{k,i}\alpha_{k,i}^{2}}{p^{2}+\alpha_{k,i}^{2}}$. Next, we need to determine the form of the incident USP, which we choose as a Gaussian form $h\left(t\right)=e^{-\gamma^{2}{\left(t-n_{0}r/c\right)}^{2}}\cos\left(\omega_{0}t-k_{0}r\right)$, where γ∝1/τ, $k_{0}=n_{0}\omega_{0}/c$.

The Gaussian momentum is chosen as one of the best known for describing USP. For example, in [28] an exact description of the subcyclic pulse beam (SCPB) was found, where, in the case considered in this paper ($\omega_{0}/\gamma \gg 1$), the solution has the form of a Gaussian pulse. In the chosen USP case, we obtain $\tilde{h}\left(\omega \right)=\frac{\sqrt{\pi }}{2\gamma }\left(e^{-{\left(\omega -\omega_{0}\right)}^{2}/4\gamma^{2}}+e^{-{\left(\omega +\omega_{0}\right)}^{2}/4\gamma^{2}}\right)$. Consider the case of multi-cycle momentum, i.e., when $\omega_{0}/\gamma \gg 1$, which is mainly used in diffraction analysis of matter. Using the expression (Equation 3), we obtain
(5)$$\begin{array}{c} \frac{d\varepsilon }{d\Omega_{k}}=\frac{{\left[E_{0}n\right]}^{2}}{8c^{3}\gamma \sqrt{2\pi }}\left[\sum_{i=1}^{A}N_{e,i}N_{A,i}\left(1-\right|F_{i}{|}^{2})+\sum_{i,j=1}^{A}\delta_{i,j}N_{e,j}F_{i}F_{j}^{*}\right] \end{array}$$

The expression in square brackets in (Equation 5) differs from the same in (Equation 3) only in that you have to replace $\omega \to \omega_{0}$. In other words, the scattering is at the incident pulse frequency $\omega_{0}$.

Next, we carry out calculations of the scattering of the USP on nucleotides using the resulting Equation (Equation 5). The Figure 2, Figure 3, Figure 4 and Figure 5 show the results of the USP scattering calculations on nucleotides: adenine, see Figure 2; guanine, see Figure 3; thymine, see Figure 4; cytosine, see Figure 5. The calculation results in all figures are normalized to the maximum value of the scattering spectrum.

The value $\omega_{0}=2c$, which corresponds to the photon energy $\hbar \omega_{0}=7.46$ keV. The choice of the USP intensity value $I\propto E_{0}^{2}$ and the duration γ do not affect the spatial distribution of the scattering intensity; therefore, these parameters may not be set (considering the chosen normalization to the maximum value of the scattering spectrum). From Figure 2, we can see that there is one main diffraction peak at scattering and many smaller ones. All in all, we can distinguish four small peaks that dominate the others. In Figure 3, there is also one main diffraction peak at scattering and many smaller ones. In general, three small peaks can be distinguished, which dominate the others.

In Figure 4, there is also one main diffraction peak at scattering and many smaller ones. In general, we can distinguish three small peaks that dominate the others, although these three peaks dominate insignificantly. In Figure 5, there is also one main diffraction peak at scattering and many smaller ones. In general, it is not possible to distinguish small peaks that clearly dominate the others. The directions of the smaller peaks in all of the figures are complex and are set by the spatial arrangement of the atoms in a given molecule. The arrangement of the small peaks is asymmetric, which is due to the asymmetric arrangement of the atoms in the nucleotides. One can also see that, in general, the direction and size of most of the scattered peaks are located in the direction of the incident pulse.

## 4. Discussion and Conclusions

Thus, we obtained a general Equation (Equation 3) for calculations of scattering spectra of USP on complex polyatomic structures. The main value responsible for the spatial arrangement of atoms in the system is determined by the parameter $\delta_{i,j}$, which is calculated by Equation (Equation 4). In the case of multi-cycle USP, Equation (Equation 3) can be represented as (Equation 5). The obtained expressions have an analytical form, which greatly simplifies the calculations and the interpretation of the results obtained. As examples, we considered scattering on DNA nucleotides: adenine, guanine, thymine, and cytosine.

Scattering on nucleotides has its own diffraction pattern by which one can judge the structure of a nucleotide. The results obtained have good prospects for the further investigation of scattering spectra of complex polyatomic structures, including those for complex biomolecules. These biomolecules include high molecular weight organic substances, for example, proteins, etc. Indeed, proteins consist of alpha-amino acids linked in a chain by peptide bonds, which makes it possible to reveal a certain pattern and find the $\delta_{i,j}$ factor. Such complex structures include various modern composite materials and nanostructures.

Of particular interest is the study of promising materials for quantum technologies, for example, quantum bits at room temperature in two- and three-dimensional solids. The resulting Equation (Equation 3) is more general and relative to the well-known expression in diffraction analysis theory [6]. Equation (Equation 3) takes into account the characteristics of the USP: its duration and shape, as well as the incoherent part in the scattering spectrum. From Equation (Equation 3) in the frequent case, we can obtain the well-known expression in diffraction analysis if we do not take into account the incoherent part in the scattering spectrum and if we consider the duration of the USP to be infinitely long, so that the condition $\omega_{0}/\gamma \gg 1$ is satisfied—i.e., the pulse must be multicyclic.

It is necessary to use Equation (Equation 3) for systems consisting of a small number of atoms, since the incoherent part in the scattering spectrum makes a significant contribution. This is easily shown using Equation (Equation 5) at $\lambda_{0}\ll 1$, and then ${\left(\frac{d\varepsilon }{d\Omega_{k}}\right)}_{max}∼N_{e}+N_{e}^{2}{\left(\lambda_{0}/a\right)}^{4}$, where a∼1, and $N_{e}$ is the total number of electrons in the polyatomic structures. This is an important refinement because the incoherent part in the scattering spectra of X-ray USPs is usually not taken into account. For polyatomic structures and low-cycle USPs, the contributions of the incoherent and coherent components may also be comparable. In general, Equation (Equation 3) should always be used for low-cycle pulses, since the shape and duration of the pulse are important.

## Figures and Tables

**Figure 1 ijms-23-00163-f001:**
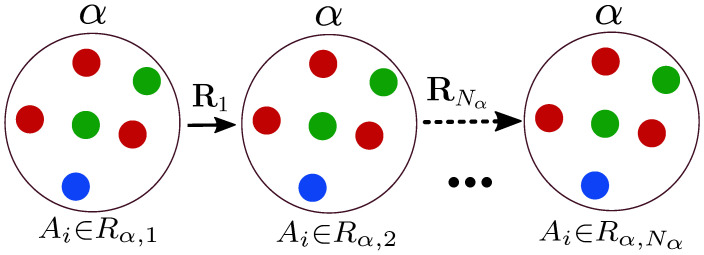
Schematic representation of the main parameters included in the factor $\delta_{i,j}$ calculated by the Equation (Equation 4). Multicolored circles are atoms; one color specifies a certain kind of atoms. If the arrangement of atoms in the system is repeated—first, second, etc. up to $N_{\alpha }$ large circles, this sets the symmetry α, and $R_{1},R_{2},\cdots R_{n_{\alpha }},\cdots ,R_{N_{\alpha }}$ are radius vectors setting the position of $1,2,\cdots ,n_{\alpha },\cdots ,N_{\alpha }$ large circles. For example, the symmetry with α=1 is represented in this figure, and with α=2 the location, color, number of circles inside the big circle, number of big circles, and $R_{n_{2}}$ would be different.

**Figure 2 ijms-23-00163-f002:**
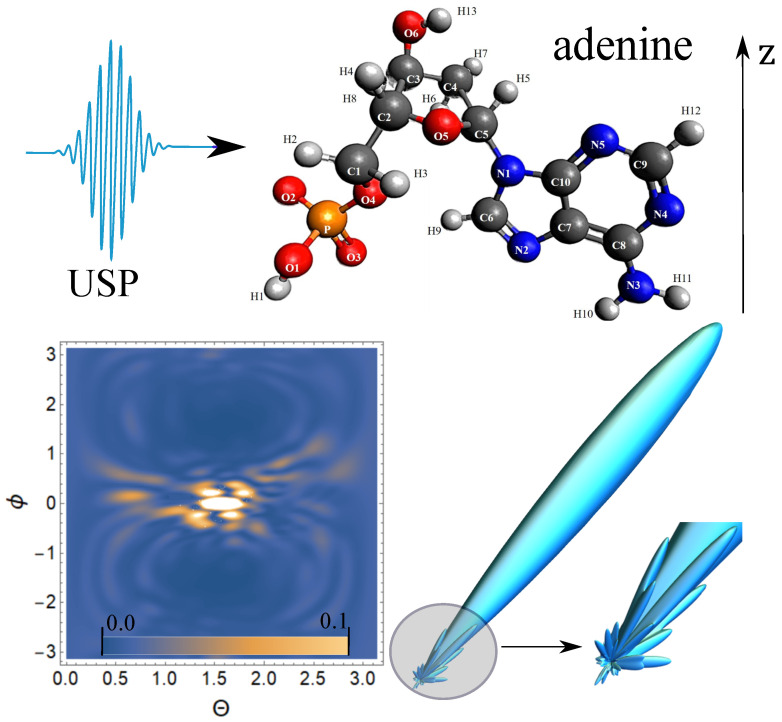
The spectra of USP scattering on adenine are presented. The direction of incidence of the USP corresponds to the representation in the figure and is perpendicular to the *z* axis. A contour plot of the normalized scattering spectrum is shown on the bottom left, and a 3D spatial scattering spectrum with a clipping from the region where the scattering is most diverse is shown on the bottom right.

**Figure 3 ijms-23-00163-f003:**
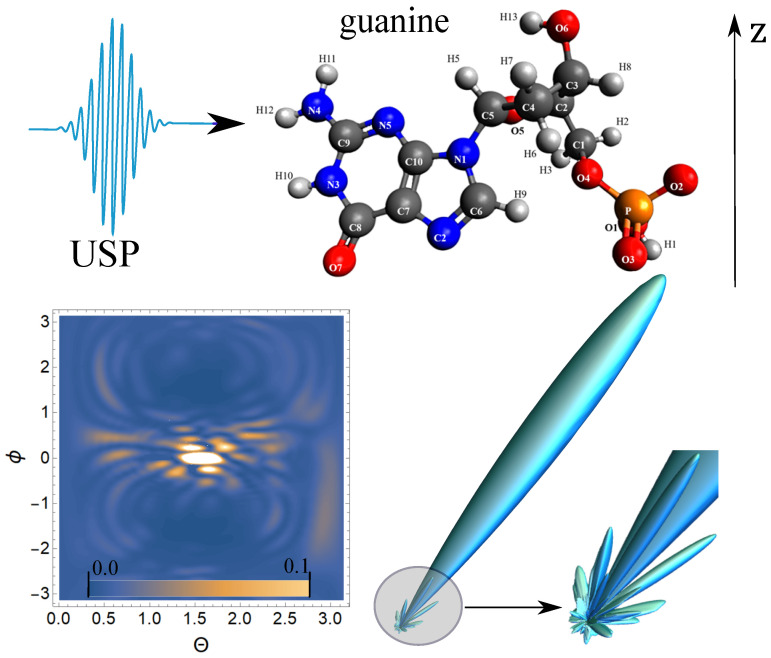
The spectra of USP scattering on guanine are presented. The rest is similar to Figure 2.

**Figure 4 ijms-23-00163-f004:**
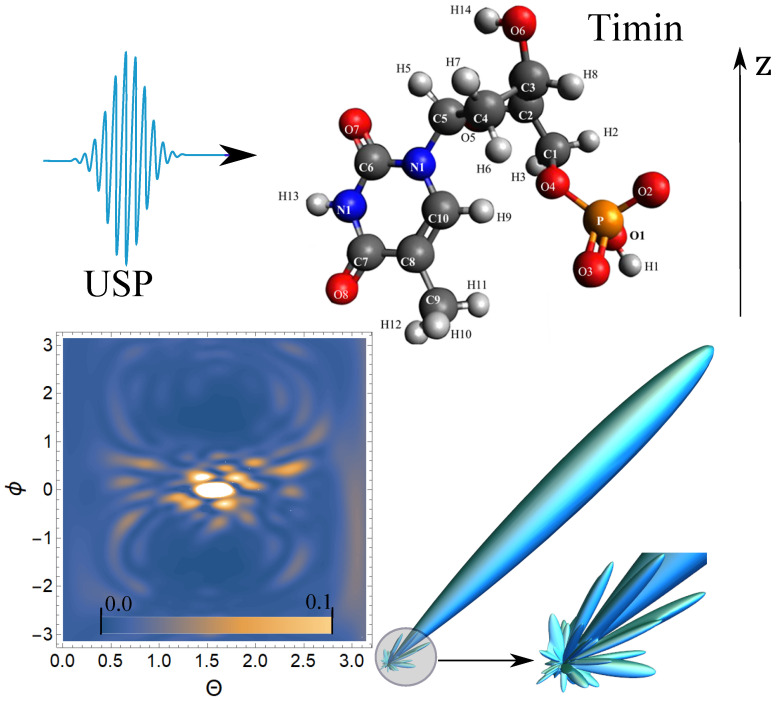
The spectra of USP scattering on thymine are represented. The rest is similar to Figure 2.

**Figure 5 ijms-23-00163-f005:**
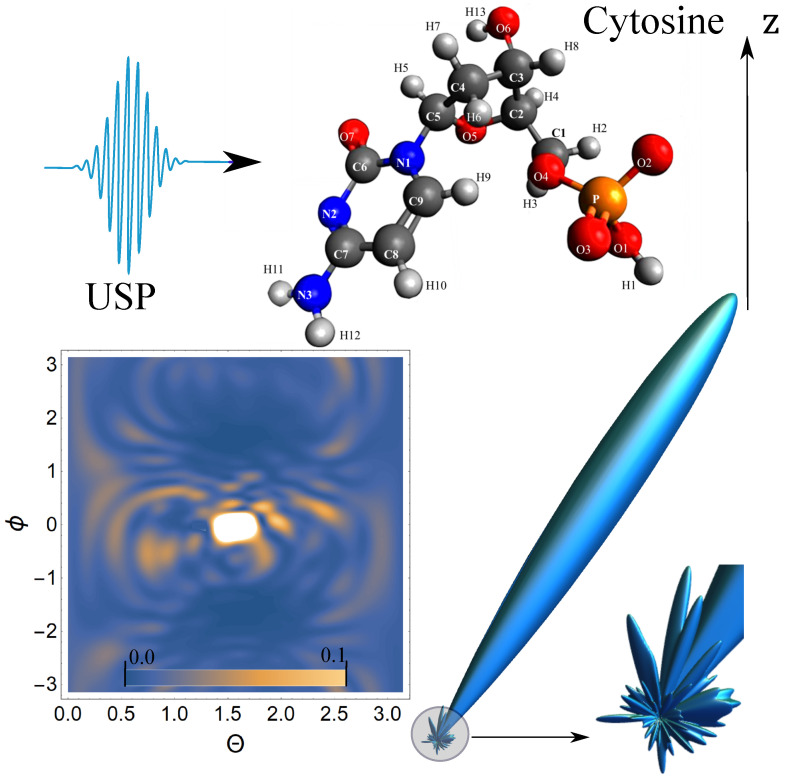
The scattering spectra of USP on cytosine are presented. The rest is similar to Figure 2.

## Data Availability

Request data from the corresponding author of this article.

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
