# Peer review of "Scattering of X-ray Ultrashort Pulses by Complex Polyatomic Structures"

_ijms, 2021, doi:10.3390/ijms23010163_

Round 1
Reviewer 1 Report
In this article, the authors have performed theoretical studies on the scattering of X-ray ultrashort pulses by biological macromolecules. To verify the theoretical basis proposed in this study, they have used DNA bases A, T, G, and C. In my opinion, this is a good example of developing new knowledge about the less studied interactions of ultrashort pulses with complex polyatomic structures. The results of the current study may also provide a good perspective for further study of scattering spectra of complex polyatomic biomolecules. I would suggest the authors to add a discussion on how the results of the current study can be applied to other biomolecules such as proteins. Including a paragraph to this effect in the Discussion and Conclusion section would be beneficial to a broader readership.
Another suggestion would be to rewrite a part of the introduction section. For example, the text in lines 32-33 and 51-53 needs to be reordered because it was copied from authors another recent publication (M K Eseev et al. 2021, Scientific Reports). Out of concern for academic integrity and to avoid future ethical problems, I would suggest that the authors rewrite this introductory section.
Author Response
Authors' responses:
Added in Discussion and Conclusion section:
“These biomolecules include high molecular weight organic substances, for example, proteins, etc. Indeed, proteins consist of alpha-amino acids linked in a chain by a peptide bond, which makes it possible to reveal a certain pattern and find the δi,j factor. Also, such complex structures include various modern composite materials and nanostructures. Of particular interest is the study of promising materials for quantum technologies, for example, quantum bits at room temperature in two and three-dimensional solids.”
Changes have been made to two sentences indicated by the reviewer:
“The theory of X-ray diffraction is usually constructed in the approximation of plane wave scattering of infinite time duration [15].”
“We will assume that the USP falls in the direction of n0 onto a complex polyatomic structure. We will assume that the time duration of such a pulse τ is much shorter than the characteristic atomic time τa ~1, i.e. let us assume that τ<< τa.”

Reviewer 2 Report
In this manuscript, the authors develop an analytical method to calculate the scattering spectra of ultrashort pulses by polyatomic structures. This is especially important because, with the advent of powerful ultrashort electromagnetic field sources, the problem of studying the scattering of such pulses in matter is particularly relevant.
There are, however, a few minor questions and comments which arise:
- In most cases, some symbols (or variables, or parameters) shown in the different equations or in the text are not adequately defined. In some other cases, they are first introduced in the equations and then defined much later in the text. Finally, some symbols (e.g. α) are used twice for indicating different things.
- in line 66, Is this "W/sm" correct?
- The symbol "h" in Eq. (2) should be same as one in line 77 "where ...".
- The symbol "s" in Eq. (3) should be adequately defined.
- in line 108, " of coherent over coherent" should be " of coherent over incoherent"?
- in line 161, " In Fig." should be " Fig." ?
However, beyond these minor comments, I feel this paper presents enough interesting and novel content to recommend acceptance of this manuscript for publication.
Author Response
Authors' responses:
- The value of in one case is replaced by , since two values with the same symbol duplicate each other. Also, all symbols are now properly defined.
- Typo. Replaced by W/cm2.
- Yes, that's a typo. Changed in Eq. (2).
- Added to text “A is the number of varieties of atoms.” The s symbol has been replaced by A so as not to be confused with another value in the text.
- Yes, that's a typo. Changed on this line.
- Replaced by "The figures ..."
